# Dietary Protein Intake and Determinants in Māori and Non-Māori Octogenarians. Te Puāwaitanga o Ngā Tapuwae Kia Ora Tonu: Life and Living in Advanced Age: A Cohort Study in New Zealand

**DOI:** 10.3390/nu12072079

**Published:** 2020-07-14

**Authors:** Anishka Ram, Ngaire Kerse, Simon A. Moyes, Marama Muru-Lanning, Carol Wham

**Affiliations:** 1College of Health, Massey University, Auckland 0632, New Zealand; anishka.ar@gmail.com; 2School of Population Health, University of Auckland, Auckland 1072, New Zealand; n.kerse@auckland.ac.nz (N.K.); s.moyes@auckland.ac.nz (S.A.M.); 3James Henare Māori Research Centre, University of Auckland, Auckland 1142, New Zealand; m.murulanning@auckland.ac.nz

**Keywords:** older adults, protein intake, protein distribution, food sources, LiLACS NZ

## Abstract

Protein intake, food sources and distribution are important in preventing age-related loss of muscle mass and strength. The prevalence and determinants of low protein intake, food sources and mealtime distribution were examined in 214 Māori and 360 non-Māori of advanced age using two 24 h multiple pass recalls. The contribution of food groups to protein intake was assessed. Low protein intake was defined as ≤0.75 g/kg for women and ≤0.86 g/kg for men. A logistic regression model was built to explore predictors of low protein intake. A third of both women (30.9%) and men (33.3%) had a low protein intake. The main food group sources were beef/veal, fish/seafood, milk, bread though they differed by gender and ethnicity. For women and men respectively protein intake (g/meal) was lowest at breakfast (10.1 and 13.0), followed by lunch (14.5 and 17.8) and dinner (23.3 and 34.2). Being a woman (*p* = 0.003) and having depressive symptoms (*p* = 0.029) were associated with consuming less protein. In adjusted models the odds of adequate protein intake were higher in participants with their own teeth or partial dentures (*p* = 0.036). Findings highlight the prevalence of low protein intake, uneven mealtime protein distribution and importance of dentition for adequate protein intake among adults in advanced age.

## 1. Introduction

By 2051, the 85 years and over age group are expected to make up almost a quarter of the 65+ population [1]. The very old experience more co-morbidities and disabilities as they get older and can potentially place a high burden on health care costs and services [2]. We previously determined half (49%) of Māori and 38% of non-Māori octogenarians were at high risk of malnutrition [3].

Those who are malnourished may have an inadequate protein intake which could contribute to possessing lower muscle mass and strength [4]. Older adults have a lower rate of anabolism and are more likely to have poorer immune functions compared to younger adults, so their protein intake need is higher for their weight and physical activity [5]. Adequate protein intake is important for optimal muscle synthesis where protein intake below the estimated average requirements (EAR) of 0.75 g/kg/day for women and 0.86 g/kg/day for men using the New Zealand Nutrient Reference Values (NRVs) [6] can lead to decreased physical function and can contribute to sarcopenia.

Protein can be obtained from both plant and animal sources; though they differ greatly in protein quality, rate of digestion and biological value, which affects the rate of skeletal muscle protein synthesis [7]. For example, animal sources such as meat, eggs and seafood have almost all the essential amino acids, higher biological value and can be digested quicker than plant sources such as vegetables and grains [8]. In the very old, bread and milk are the main source of protein in women and beef/veal, fish and seafood and bread in men [9].

The distribution of protein intake may also influence the maintenance of lean muscle mass. A key strategy for increasing the rate of protein synthesis is to evenly spread protein throughout the day, by aiming to have between 25 to 30 g of protein in each main meal [10]. In the Newcastle 85+ study of adults in advanced age, it was observed that protein intake was skewed towards the end of the day. Protein intake was lowest in the morning especially at breakfast whereas, the largest amount of protein was consumed during the evening meal [11].

Therefore, the aim of this study was to investigate the intake, food sources, distribution, and adequacy of protein among Māori and non-Māori participants in Te Puāwaitanga o Ngā Tapuwae Kia Ora Tonu; Life and Living in Advanced Age: A Cohort Study in New Zealand.

## 2. Materials and Methods

### 2.1. Life and Living in Advanced Age: A Cohort Study in New Zealand (LiLACS NZ)

We conducted a cross-sectional study using data from the baseline and follow-up assessment of Life and Living in Advanced Age, A Cohort Study in New Zealand (LiLACS NZ), a longitudinal study of the very old in New Zealand [12]. Eligibility criteria were Māori aged 80 to 90 years and non-Māori aged 85 years living within the Bay of Plenty and Lakes District Health Board regions (excluding Taupo region). Younger Māori participants were recruited as the gap in life expectancy between Māori and non-Māori was 8.2 years for men and 8.8 years for women [13]. The details of the recruitment process are described elsewhere [14].

At baseline in 2010, 671 participants completed a comprehensive questionnaire, conducted by trained interviewers and a health assessment by a trained nurse [12]. Participants’ age, demographic, and health characteristics were established. Rates of participation varied across questionnaires and assessments.

A 12-month follow-up visit was undertaken in 2011 and included a physical assessment including weight, body mass index (BMI), fat mass and muscle mass as well as grip strength and level of physical activity. A detailed dietary assessment using the 24 h multiple pass recall (24 h MPR) on two separate days of the week was offered as part of that stage of the study.

### 2.2. Socio-Economic, Health and Physical Factors

At baseline, we established whether the participants had practical help available. Participants responded either yes/no to: “When you need some extra help, can you count on anyone to help with daily tasks like grocery shopping, cooking, house cleaning, telephoning, give you a ride?” Current living arrangement was categorized as living alone (yes/no). The NZ Deprivation (NZDep) index obtained from the Ministry of Health was used as an indication of sociodemographic deprivation. The index was constructed from geo-coded addresses and included eight dimensions of material and social deprivation reflecting lack of income, employment, communication, transport, educational qualifications, home ownership and living space [15]. During follow-up participants responded yes/no to whether they received a pension only income.

Health factors were established at baseline. Oral health was established by whether the participants wore dentures, had trouble with biting and chewing or had swallowing problems. Depression was assessed by the 15-item Geriatric Depression Screening Scale (GDS-15) [16], a reliable and valid self-rating depression screening scale developed specifically for older people [17]. A higher score indicates more depressive symptoms, with a cut-off of 5 or more considered to indicate significant depressive symptoms [18,19]. The twelve question version 2 of the Short Form health score (SF-12) [20] was used to provide a view of health related quality of life (QoL) of the participants based on their perceived experience, knowledge and awareness of their personal, physical, mental and emotional status. The scale presents two summary scores: Physical and mental health related QoL. The maximum score is 100; any score lower than 40 is indicative of perception of poor health and above 60 is indicative of reasonable and better health [20].

At follow-up the participants height was measured with a SECA 213 free-standing stadiometer two times unless the variance between both figures were greater than 1 cm then a third attempt was recorded [12]. Weight was estimated using a Tanita digital measuring scale (BC-545, Tanita Corporation) and height and weight were used to calculate BMI using the formula kg/m^2^. Measurement of fat mass and muscle mass were estimated by bioimpedance using the Tanita scale (Inner Scan Body Composition Monitor, BC-541, Tanita Corporation, Tokyo, Japan). Grip strength was estimated using a Takei digital handgrip dynamometer-Grip D. The average value of three readings from the strongest hand was recorded [12]. A Physical Activity Scale for the Elderly (PASE), validated in community living adults [21] was used to assess physical activity. PASE consists of ten items used to identify leisure, household and occupational related activity and duration of each activity over a one-week period. The total PASE score was derived by multiplying the duration of each activity (h/week) or participation (yes/no) by the empirically derived item weights and summing over all activities. Also, the following information was collected, age and pension only income was ascertained using the question “What are your sources of income?” The answers were categorized into either yes/no.

### 2.3. Dietary Assessment

As part of the follow-up assessment, 214 Māori and 360 non-Māori participants completed a 24 h MPR on two separate days of the week. The validity of the 24 h MPR is detailed elsewhere [22,23]. Interviewers were instructed to record the weight of the food either by reading food labels or estimating the portion size with household utensils. The ‘Photographic Atlas of Food Portion Sizes”, previously applied in the Newcastle 85+ study was adapted for the New Zealand diet and used to assess the portion of foods when an item could not be quantified [24,25]. Nutrient intakes were calculated by coding all foods and drinks recorded by the participants using the New Zealand Food Composition Database (NZFCDB). FOODFiles (2010), an electronic subset of data from the NZFCDB was used as the main source of food composition data and contained information on 2739 foods.

Protein intake was assessed using the NZ NRV’s [25] and estimated average requirements (EAR) in grams per kilogram adjusted for body weight per day, to determine the adequacy in men (0.86 g/kg/day) and women (0.75 g/kg/day). Protein distribution was assessed across breakfast, lunch, and dinner in grams per meal.

All foods from the 24 h MPR were allocated into one of thirty-three food items, used in the New Zealand Adult Nutrition Survey (NZANS) (2008/09) that contributed to protein intake [26]. The top fifteen food items that contributed the most protein to the participants’ diet was reported. Supplements were not included.

### 2.4. Statistical Analysis

The Kolmogorov-Smirnov and Shapiro-Wilk tests were used to check for normality for variables (age, has practical help available, living alone, socio-economic deprivation, pension only income, wears dentures, difficulty biting and chewing, swallowing difficulties, depression, SF-12 physical health, SF-12 mental health, BMI, weight, grip strength, PASE score, fat mass percentage and muscle mass percentage). Categorical data were presented as percentage with corresponding sample size. Normally distributed data were expressed as a mean and standard deviation (±SD); non-Gaussian distributed variables were expressed as medians and interquartile ranges (IQR). The Mann-Whitney *u* test and independent *t*-test were used to determine differences between Māori and non-Māori, men and women for age, SF-12 physical health, SF-12 mental health, BMI, weight, grip strength, PASE score, fat mass percentage, muscle mass percentage, energy, protein (energy %), protein (g/day), protein (g/kg/day), protein distribution (breakfast, lunch and dinner) and chi-square tests were used to test differences for categorical data (practical help available, living alone, socio-economic deprivation, pension only income, wears dentures, difficulty biting and chewing, swallowing difficulties, depression, total protein, acceptable macronutrient distribution range (AMDR), and low and adequate protein based on EAR). Protein from food groups were presented as a mean (±SD) and contribution to protein intake was expressed as a percentage (%) and adjusted for body weight (g/kg/day).

A reduced multivariate regression model was built to predict protein intake in g/kg/day. Age, gender and ethnicity were included in the model and other potential predictors included practical support, depression, living alone, NZDep (score), wears dentures, chewing problems, swallowing difficulties, SF-12 physical and mental health scores, and pension only income. Variables with the highest *p*-value were removed in the model and *p* ≤ 0.2 was considered statistically significant enough to warrant inclusion. A logistic regression model was constructed to predict meeting the EAR for adequate protein intake (g/kg/day). The IBM SPSS Statistics 23 program was used to conduct the statistical analysis.

### 2.5. Ethics Approval

The study was granted an ethics approval by the Northern Regional Ethics Committee of New Zealand in December 2009 (NTX/09/09/088).

## 3. Results

### 3.1. Participants Characteristics

At baseline 267 Māori and 414 non-Māori completed all measures. The sociodemographic health and physical characteristics of Māori and non-Māori men and women are illustrated in Table 1. The median age for Māori was 83 years (IQR 81–85), slightly younger than non-Māori with a median age of 86 years (IQR 85–86) (*p* < 0.001). More women (51% Māori, 65% non-Māori) than men (25% Māori, 36% non-Māori) lived alone (*p* < 0.001). Also, more women (54% Māori, 54% non-Māori) had full mouth dentures compared to men (42% Māori, 40% non-Māori) (*p* = 0.011). Rates of participation varied across the assessments both at baseline and at 12 months follow-up.

### 3.2. Dietary Assessment 24 h Multiple Pass Recall (MPR)

At 12 months follow up 224 Māori and 370 non-Māori completed all interview measures. There were 214 (96%) Māori and 360 (97%) non-Māori participants that completed the 2 × 24 h MPR dietary assessment and were weighed. Table 2 shows the energy, protein intake, distribution and requirements for Māori and non-Māori men and women. Women (1466 kcal/day (IQR 1219–1755)) were more likely to consume less total energy than men (1875 kcal/day (IQR 1537–2247)) (*p* < 0.001). Protein intake per day per kilogram of body weight was also significantly lower for women (58.6 g/day), (0.9 g/kg/day) compared to men (74.6 g/day), (1.0 g/kg/day) (*p* ≤ 0.001). Furthermore, women were more likely to consume less protein (g/meal) than men; at breakfast (10.1 g, 13.0 g), lunch (14.5 g, 17.8 g) and dinner (23.3 g, 34.2 g) (*p* < 0.001), respectively. Protein intake was greatest during the evening meal. On average women and men consumed 15.5% and 15.8% of their energy from protein, respectively (*p* = 0.003). Most participants met the EAR for protein intake for both women (69%) and men (67%). Māori (66% women, 72% men) were more likely to meet the AMDR for protein than non-Māori (56% women, 56% men) (*p* = 0.003).

### 3.3. Top Food Group Contributors to Protein Intake

As reported previously [8], % contribution from food groups to protein intake is shown in Table 3 and Table 4. Food groups are ranked from high to low by adequate protein intake (g/kg/day). Consumption of adequate and inadequate protein (g/kg/day) and protein intake (%) of the top 15 food groups based on EAR for women and men are shown. Overall, the largest contributors of adequate protein intake, for Māori and non-Māori were beef/veal, bread, and milk.

### 3.4. Determinants of Protein Intake (g/kg/day)

Controlling for age and ethnicity, a reduced multivariate regression model was built to predict determinants of protein intake (Table 5). For Māori and non-Māori participants being a woman (*p* = 0.003) and having depressive symptoms (*p* = 0.029) was associated with consuming less protein.

### 3.5. Logistic Regression Model to Predict Meeting the EAR for Protein (g/kg/day)

Table 6 shows a logistic regression model to predict meeting the EAR for protein in grams per kilogram per day for women (≤0.75 g/kg/day) and men (≤0.86 g/kg/day), after controlling for age and ethnicity. Māori and non-Māori participants with no dentures (their own teeth) or partial dentures were more likely to meet the EAR (*p* = 0.036).

Protein intake was not associated with health related QoL or socio-economic deprivation in either analysis.

## 4. Discussion

### 4.1. Protein Intake and Adequacy

Overall, we found the median protein intakes were 0.9 g/kg/day for women and 1.0 g/kg/day for men. A third of participants (30.9% of women, 33.3% of men), did not meet the EAR for older adults ≥70 years in Australia and New Zealand (≤0.75 g/kg/day women, ≤0.86 g/kg/day men) [6]. Protein intakes were similar to findings among adults (85 years) in the Newcastle 85+ study where the median protein intake was 0.96 g/kg/day women and 1.04 g/kg/day for men [11].

In Canada, among participants aged from 67 to 84 years in the NuAge study, the mean protein intake for men was 1.05 g/kg/day and 1.04 g/kg/day for women) [27] and among Dutch community dwelling older adults (mean age 77 years) men and women were observed to consume on average 1.07 g/kg/day and 1.05 g/kg/day of protein respectively; 10% of all participants did not meet the RDA (>0.8 g/kg/day) [28].

The optimal level of protein intake remains to be determined especially for those in advanced age. Australian and New Zealand recommendations for protein intake are aggregated for adults over 70 years and based on data derived from nitrogen balance studies in young adult men [6]. The EAR for adults >70 years was increased by 25% over that of young adults [29] without robust data on which to base that estimate. An international study group for optimal dietary protein intake in older people (PROT-AGE) and the European Society for Clinical Nutrition and Metabolism (ESPEN) suggest protein requirements need to be increased to 1.0–1.2 g/kg/day for healthy adults over 65 years to overcome anabolic resistance, sarcopenia and reduce the loss of muscle mass [6,30].

Using a reduced regression model adjusted for age and ethnicity we found being a woman (*p* = 0.003) and the presence of depressive symptoms (*p* = 0.029) predicted a lower protein intake adjusted for body weight. Conversely in the Newcastle 85+ study being a woman was associated with adequate protein intake (≥0.8 g/kg/day) in adjusted models [11].

There are a few studies which suggest that protein intake is higher in men compared to women adjusted for body weight. Among Brazilian older adults aged 60 to 104 years, a higher average protein intake (1.18 g/kg/day) was observed in men compared to women (1.01 g/kg/day) [31]. Similarly, among free-living Japanese older adults (≥70 years) women consumed significantly less protein (1.5 g/kg/day) than men (1.8 g/kg/day) [32]. Older women are reported to consume lower quality protein across the day and eat less meat or meat products as a result of being more health conscious than men [33]. Among older women who lived alone in the UK, an exploratory study found that the need to cook high protein foods, their often perishable nature and high cost were barriers to consuming these foods [34] and this may be a contributing factor in the current study as more women lived alone than men.

Our findings observed that the presence of depressive symptoms was associated with consuming less protein as had been observed among older men (>70 years) in South Korea; those with an inadequate protein intake were almost nine times more likely to be at risk of having depression [35]. It has also been observed that depressive symptoms were significantly increased among those who consumed a low protein diet (<0.8 g/kg/day) among older Italians with type 2 diabetes and chronic renal disease [36]. Depression has an impact on appetite and overall food intake [37] and in the baseline assessment of LiLACS NZ we found depressive symptoms were associated with high nutrition risk for non-Māori [3].

Using a logistic regression model controlled for age and ethnicity we found the oral health of the participants predicted meeting the EAR for protein for both women (≥0.75 g/kg/day) and men (≥0.86 g/kg/day). Participants with their own teeth or partial dentures were more likely to have an adequate protein intake than those with full dentures (*p* = 0.036). Edentulism was more common in women (54%) than men (41%) which may partially account for the lower protein intake (g/kg/day) observed among women. In New Zealand removing all teeth at a young age was common practice during the early to mid-twentieth century and at this time New Zealand had the highest rate of edentulous people in a developed country [38]. Inadequate access to dentists and lack of income for on-going dental care led to an ‘edentulism epidemic’ with individuals often being between the ages of 15 to 40 years at the time of extraction [38]. In the Newcastle 85+ study, it was observed that a higher protein intake (RDA > 0.8 g/kg/day) was significantly associated with a higher tooth count [11] which confirms the importance of oral health for eating foods which provide an adequate protein intake. Similarly, among individuals aged 74 years in Japan, those with ≥21 versus <20 teeth had a significantly higher intake of total protein and animal protein [39] and in Bulgaria individuals aged 47 to 89 years with less teeth also had a significantly lower intake of protein and animal protein (meats) [40].

Older adults with no teeth or ill-fitting dentures are known to have more difficulties in chewing hard textured foods such as meat [41] and denture wearers have a decreased bite force and experience pain compared to those with natural teeth [42] all of which may compromise protein intake. Our study adds to the body of evidence that the oral health status of adults in advanced age is generally deficient, and surveillance and improvement of oral health in older people should be a key priority.

### 4.2. Protein Distribution

We observed a skewed protein distribution (g/meal), with the highest intake at dinner and the least at breakfast. This may reflect the typical New Zealand eating pattern where protein is eaten mostly at the dinner meal. In the Newcastle 85+ study, 22% of the protein intake occurred in the morning, most (35%) during the “lunch” period followed by 21% over the “dinner” period [11]. The latter observation may have led to earlier satiety in the day resulting in a lower protein intake at dinner [43]. In the Netherlands, protein intake has also been observed to be lowest at breakfast (9.9 g), and highest at both lunch (28.0 g) and dinner (27.0 g) among community dwelling older adults (75 to 97 years) [28]. Likewise, older men and women (67 to 82 years) had a lower protein intake at breakfast (16 g, 11 g) and similar intakes across lunch (28 g, 26 g) and dinner (32 g, 26.5 g), respectively in the NuAge study [44]. Similarly, among German community dwelling older adults (75 to 85 years) the least amount of protein was consumed at breakfast (16.5g), with slightly more at lunch (24.2 g) and dinner (21.9 g) [45]. As protein intakes of 25 to 30 g spread evenly across breakfast, lunch and dinner have been suggested to stimulate muscle protein synthesis [4,46] and lead to increased muscle mass retention [44], our findings suggest that a food-based approach for increasing protein intake needs to include increased protein at the breakfast meal.

### 4.3. Protein Sources

For participants with an adequate protein intake (>EAR), beef/veal, bread and milk were the top three sources of protein and sources differed between sex and ethnic groups. In those with a low protein intake (≤EAR) bread was the main source followed by milk and beef/veal for Māori women and non-Māori men and women. This differed for Māori men where beef/veal was the main source followed by bread and fish and seafood. For Māori, fish and seafood (kāimoana) are traditional kai (foods) and hold a strong importance in Māori well-being [47] especially where the marae (meeting house) is closer to the sea [48]. However, due to the shortage of kāimoana as a result of pollution, Māori predominantly consume a Western diet that includes food such as bread and milk that are lower in protein content as reflected in the Māori octogenarians in this study.

Our findings are consistent with the NZANS survey (2008/09) which observed the largest food contributors of protein were bread (14.3%, 14.2%), followed by milk (10.8%, 11.5%) and beef/veal (10.1%, 9.3%) in both men and women aged over 70 years, respectively [26].

### 4.4. Strengths and Limitations

Our study is the first to report the determinants of protein intake among Māori and non-Māori octogenarians in New Zealand using a repeat 24 h multiple pass recall which provides the best available method in this age group [22]. Māori participants were younger, with a higher BMI, than non-Māori and culturally different food sources of protein were evident. Mis-reporting and under-reporting may have occurred. However, protein rich foods are usually not understated compared to food items with a negative health image (e.g., cakes, sweets, confectionery) [49]. Detailed comparisons between studies may be hindered by different dietary assessment methods and participants characteristics such as geographic location, age, body composition as well as health and nutritional status.

## 5. Conclusions

This study found a third of the participants did not meet the EAR for adequate protein intake. We observed an uneven mealtime protein distribution pattern, and the food group contributors to protein intake differed by gender and ethnicity. Depression was associated with protein intake, and the odds of consuming an adequate protein intake were higher in participants with their own teeth or partial dentures (adjusted models). Our findings highlight the prevalence and determinants of low protein intake among Māori and non-Māori of advanced age which may help to inform preventative interventions to improve the nutritional health of the oldest old.

## Figures and Tables

**Table 1 nutrients-12-02079-t001:** Sociodemographic, health and physical characteristics of Māori and non-Māori men and women.

	Māori			Non-Māori			*p*-Value ¥	*p*-Value ◊
	Women	Men	Total Māori	Women	Men	Total Non-Māori	(Ethnic Group)	(Sex)
	*n* (%)	*n* (%)	*n* (%)	*n* (%)	*n* (%)	*n* (%)		
Number	124 (57.9)	90 (42.1)	214 (37.3)	189 (52.5)	171 (47.5)	360 (62.7)		
Age (years) (Median, IQR)	84 (81–86)	82 (81–85)	83 (81–85)	86 (85–86)	86 (85–86)	86 (85–86)	*p* < 0.001*	0.452
Has practical help available								
No Yes	7 (6.6) 99 (93.4)	3 (4) 72 (96)	10 (5.5) 171 (94.5)	21 (11.5) 162 (88.5)	16 (9.5) 152 (90.5)	37 (10.5) 314 (89.5)	0.053	0.449
Living alone								
No Yes	52 (49.1) 54 (50.9)	56 (74.7) 19 (25.3)	108 (59.7) 73 (40.3)	64 (35) 119 (65)	106 (63.9) 60 (36.1)	170 (48.7) 179 (51.3)	0.017	*p* < 0.001*
Socio-economic deprivation (NZDep score)								
1–4 (least) 5–7 8–10 (most)	18 (14.5) 24 (19.4) 82 (66.1)	13 (14.4) 26 (28.9) 51 (56.7)	31 (14.5) 50 (23.4) 133 (62.1)	36 (19) 77 (40.7) 76 (40.2)	50 (29.2) 70 (40.9) 51 (29.8)	86 (23.9) 147 (40.8) 127 (35.3)	*p* < 0.001*	0.016
Pension only income								
No Yes	54 (57.4) 40 (42.6)	34 (51.5) 32 (48.5)	88 (55.0) 72 (45.0)	118 (69.4) 52 (30.6)	112 (74.7) 38 (25.3)	230 (71.9) 90 (28.1)	*p* < 0.001*	0.574
Wears dentures								
No dentures Partial Full mouth	18 (18.9) 26 (27.4) 51 (53.7)	17 (26.6) 20 (31.3) 27 (42.2)	35 (22.0) 46 (28.9) 78 (49.1)	32 (18.9) 45 (26.6) 92 (54.4)	37 (24.5) 54 (35.8) 60 (39.7)	69 (21.6) 99 (30.9) 152 (47.5)	0.903	0.011
Difficulty biting and chewing								
Never, rarely Sometimes, often, always	76 (80) 19 (20)	52 (81.3) 12 (18.8)	128 (80.5) 31 (19.5)	137 (80.6) 33 (19.4)	127 (84.7) 23 (15.3)	264 (82.5) 56 (17.5)	0.593	0.356
Swallowing difficulties								
Never, rarely Sometimes, often, always	76 (80.9) 18 (19.1)	53 (91.4) 5 (8.6)	129 (84.9) 23 (15.1)	135 (88.8) 17 (11.2)	136 (90.1) 15 (9.9)	271 (89.4) 32 (10.6)	0.158	0.129
Depression (GDS-15)								
0–4 (Not depressed) 5–15 (Depressed)	87 (82.1) 19 (17.9)	51 (68) 24 (32)	138 (76.2) 43 (23.8)	149 (81.9) 33 (18.1)	146 (86.9) 22 (13.1)	295 (84.3) 55 (15.7)	0.024	0.796
	Median (IQR)	Median (IQR)	Median (IQR)	Median (IQR)	Median (IQR)	Median (IQR)		
SF-12 Physical health	45.2 (37.0, 52.5)	46.8 (32.3, 52.4)	45.6 (35.4, 52.5)	43.0 (30.3, 51.4)	46.7 (38.1, 52.3)	44.8 (34.8, 52.1)	0.548	0.093
SF-12 Mental health	55.3 (47.7, 59.0)	54.5 (45.6, 58.9)	54.7 (47.1, 59.0)	57.1 (51.6, 60.7)	56.3 (51.7, 59.2)	56.7 (51.7, 60.1)	0.024	0.325
Anthropometry-Physical assessment								
BMI (kg/m2)	28.7 (24.0, 31.6)	28.0 (25.5, 32.1)	28.5 (24.7, 31.8)	26.5 (23.7, 30.2)	26.2 (24.1, 28.5)	26.3 (24.0, 29.3)	*p* < 0.001*	0.992
Weight (kg)	66.9 (58.8, 79.9)	77.9 (68.7, 87.6)	71.9 (63.4, 84.5)	63.8 (57.4, 72.1)	75.1 (70.0, 82.2)	70.9 (61.9, 78.4)	0.024	*p* < 0.001*
Grip strength (kg)	20.0 (17.4, 23.0)	29.9 (25.1, 35.0)	22.6 (18.3. 28.7)	18.4 (15.1, 21.3)	30.4 (25.6, 33.9)	22.6 (17.5, 30.0)	0.866	*P* < 0.001*
PASE score	77 (34, 124)	96 (51, 144)	83 (47, 138)	70 (36, 112)	86 (39, 127)	75 (36, 119)	0.109	0.012
	Mean (SD)	Mean (SD)	Mean (SD)	Mean (SD)	Mean (SD)	Mean (SD)		
Fat mass (%)	37.6 (±7.3)	29.5 (±10.1)	34.4 (±9.4)	37.9 (±6.1)	28.0 (±6.1)	33.0 (±7.9)	0.100	0.806
Muscle mass (%)	59.0 (±7.0)	67.7 (±7.4)	62.5 (±8.3)	58.4 (±7.6)	68.4 (±5.8)	63.3 (±8.4)	0.612	0.233

Variables are from Waves 1 or 2 questionnaires. Number (percentage, %). Median (25th–75th percentiles). Mean (±SD). ¥: Differences between Māori and non-Māori participants (Mann-Whitney *u* Test. Chi-square Test, Independent samples *t*-test). ◊: Differences between men and women (Mann-Whitney *u* Test, Chi-square Test, Independent samples *t*-test) * *p*-value < 0.001 considered significant.

**Table 2 nutrients-12-02079-t002:** Energy, protein intake, distribution and requirements for Māori and non-Māori men and women.

	Women			Men			*p*-Value ¥	*p*-Value ◊
	Maori	Non-Māori	Total	Maori	Non-Māori	Total	(Ethnic Group)	(Sex)
(*n*, %)	124 (39.6%)	189 (60.4%)	313 (54.5%)	90 (34.5%)	171 (65.5%)	261 (45.5%)		
	Median (IQR)	Median (IQR)	Median (IQR)	Median (IQR)	Median (IQR)	Median (IQR)		
Energy (kcal)	1433 (1147, 1724)	1499 (1267, 1793)	1466 (1219, 1755)	1747 (1444, 2164)	1890 (1604, 2295)	1875 (1537, 2247)	0.001	*p* < 0.001*
Protein (% energy)	16.2 (13.8, 19.8)	15.3 (13.4, 17.7)	15.5 (13.6, 18.1)	16.3 (14.6, 18.7)	15.5 (13.3, 17.8)	15.8 (13.6, 17.9)	0.003	0.709
Total protein AMDR (% energy)	*n* (%)	*n* (%)	*n* (%)	*n* (%)	*n* (%)	*n* (%)		
≤14% 15% and above	42 (33.9) 82 (66.1)	84 (44.4) 105 (55.6)	126 (40.3) 187 (59.7)	26 (28.9) 64 (71.1)	76 (44.4) 95 (55.6)	102 (39.1) 159 (60.9)	0.003	0.774
	Median (IQR)	Median (IQR)	Median (IQR)	Median (IQR)	Median (IQR)	Median (IQR)		
Protein (g)	55.4 (46.2, 72.3)	59.8 (48.0, 68.7)	58.6 (47.0, 70.8)	72.9 (53.4, 93.5)	75.5 (61.9, 88.0)	74.6 (59.6, 89.7)	0.270	*p* < 0.001*
Protein (g/kg)	0.86 (0.64–1.11)	0.90 (0.73–1.12)	0.87 (0.68–1.12)	0.97 (0.73–1.35)	0.97 (0.84–1.18)	0.97 (0.78–1.21)	0.176	0.001
	*n* (%)	*n* (%)	*n* (%)	*n* (%)	*n* (%)	*n* (%)		
Protein distribution (g/meal)	Median (IQR)	Median (IQR)	Median (IQR)	Median (IQR)	Median (IQR)	Median (IQR)		
Breakfast	9.7 (7.2, 14.4)	10.4 (7.0, 13.3)	10.1 (7.1, 13.7)	11.9 (8.7, 17.9)	13.1 (10.2, 17.9)	13.0 (9.4, 17.9)	0.324	*p* < 0.001*
Lunch	14.6 (9.3, 23.4)	14.4 (10.4, 20.9)	14.5 (10.0, 21.6)	16.8 (10.7, 24.5)	18.3 (12.9, 25.1)	17.8 (12.2, 25.0)	0.381	*p* < 0.001*
Dinner	23.3 (13.7, 30.3)	23.2 (14.3, 32.6)	23.3 (14.2, 31.4)	35.2 (20.1, 54.1)	33.5 (21.9, 45.6)	34.2 (21.0, 46.4)	0.873	*p* < 0.001*
	*n* (%)	*n* (%)	*n* (%)	*n* (%)	*n* (%)	*n* (%)		
¹ EAR Low Protein								
Women ≤ 0.75 g/kg Men ≤ 0.86 g/kg	42 (36.2) -	50 (27.5) -	92 (30.9) -	- 36 (42.9)	- 48 (28.6)	- 84 (33.3)	0.015	
¹ EAR Adequate Protein								
Women > 0.75 g/kg Men > 0.86 g/kg	74 (63.8) -	132 (72.5) -	206 (69.1) -	- 48 (57.1)	- 120 (71.4)	- 168 (66.7)	0.037	

Variables are from Wave 2 questionnaire. Median (25th–75th percentile). Number (*n*) (percentage, %). ¹ Low protein and Adequate protein based on Estimated Average Requirements (EAR), for over 70 years old Women 0.75 g/kg/day, Men 0.86 g/kg/day. Differences between Māori and non- Māori participants (Mann-Whitney *u*-Test, Chi-square Test, Independent samples *t*-test). Differences between men and women (Mann-Whitney *u*-Test, Chi-square Test, Independent samples *t*-test). * *p*-value < 0.001 considered significant. ¥: Differences between Māori and non-Māori participants (Mann-Whitney *u* Test. Chi-square Test, Independent samples *t*-test). ◊: Differences between men and women (Mann-Whitney *u* Test, Chi-square Test, Independent samples *t*-test).

**Table 3 nutrients-12-02079-t003:** Top 15 food group intake for Māori and non-Māori women low and adequate protein intake based on the Estimated Average Requirement (EAR).

Māori Women					Non-Māori Women				
	Contribution to Protein Intake (%)		Protein Intake (g/kg/day)			Contribution to Protein Intake (%)		Protein Intake (g/kg/day)	
Food Groups	Low protein (≤0.74 g/kg/day)	Adequate protein (>0.75 g/kg/day)	Low protein (≤0.74 g/kg/day)	Adequate protein (>0.75 g/kg/day)	Food Groups	Low protein (≤0.74 g/kg/day)	Adequate protein (>0.75 g/kg/day)	Low protein (≤0.74 g/kg/day)	Adequate protein (>0.75 g/kg/day)
	Mean (SD)	Mean (SD)	Mean (SD)	Mean (SD)		Mean (SD)	Mean (SD)	Mean (SD)	Mean (SD)
Beef, veal	7.3 (±12.5)	12.9 (±15.9)	0.07 (±0.13)	0.13 (±0.16)	Milk	12.7 (±12.4)	12.2 (±8.3)	0.13 (±0.12)	0.12 (±0.08)
Milk	10.1 (±7.9)	11.0 (±8.0)	0.10 (±0.08)	0.11 (±0.08)	Bread	13.5 (±9.2)	11.3 (±6.8)	0.14 (±0.09)	0.11 (±0.07)
Bread	17.4 (±7.6)	10.5 (±6.7)	0.17 (±0.08)	0.11 (±0.07)	Beef, veal	9.4 (±11.6)	10.0 (±12.4)	0.09 (±0.12)	0.10 (±0.12)
Fish, seafood	6.7 (±17.0)	9.3 (±12.5)	0.07 (±0.17)	0.09 (±0.13)	Poultry	6.0 (±10.9)	6.6 (±11.8)	0.06 (±0.11)	0.07 (±0.12)
Pork	4.9 (±8.2)	9.0 (±15.5)	0.05 (±0.08)	0.09 (±0.16)	Fish, seafood	3.5 (±10.7)	5.5 (±11.7)	0.03 (±0.11)	0.06 (±0.12)
Poultry	5.8 (±9.7)	7.5 (±12.3)	0.06 (±0.10)	0.08 (±0.12)	Pork	3.0 (±6.2)	5.0 (±8.1)	0.03 (±0.06)	0.05 (±0.08)
Cereals	4.9 (±5.0)	4.6 (±3.4)	0.05 (±0.05)	0.05 (±0.03)	Cheese	4.3 (±7.8)	5.0 (±6.0)	0.04 (±0.08)	0.05 (±0.06)
Egg, egg dishes	3.7 (±7.0)	3.8 (±5.6)	0.04 (±0.07)	0.04 (±0.06)	Cereals	4.0 (±4.0)	5.0 (±4.6)	0.04 (±0.04)	0.05 (±0.05)
Vegetables	4.5 (±3.2)	3.7 (±2.2)	0.05 (±0.03)	0.04 (±0.02)	Vegetables	4.8 (±3.1)	4.8 (±3.3)	0.05 (±0.03)	0.05 (±0.03)
Cheese	3.3 (±7.9)	3.6 (±5.9)	0.03 (±0.08)	0.04 (±0.06)	Egg, egg dishes	3.2 (±5.9)	3.9 (±5.8)	0.03 (±0.06)	0.04 (±0.06)
Potato, kumara, taro	2.5 (±2.4)	3.0 (±1.9)	0.03 (±0.02)	0.03 (±0.02)	Mutton (Sheep, lamb)	0.0 (±0.0)	3.6 (±9.6)	0.00 (±0.00)	0.04 (±0.10)
Mutton (Sheep, lamb)	1.5 (±4.8)	2.6 (±9.2)	0.02 (±0.05)	0.03 (±0.09)	Fruit	3.3 (±2.9)	2.8 (±2.1)	0.03 (±0.03)	0.03 (±0.02)
Fruit	3.0 (±2.7)	2.0 (±2.0)	0.03 (±0.03)	0.02 (±0.02)	Sausage, processed meats	2.7 (±7.3)	2.3 (±5.6)	0.03 (±0.07)	0.02 (±0.06)
Sausage, processed meats	3.0 (±7.0)	1.7 (±4.7)	0.03 (±0.07)	0.02 (±0.05)	Cakes	3.1 (±4.3)	2.3 (±3.3)	0.03 (±0.04)	0.02 (±0.03)
Non-alcoholic beverages	2.3 (±1.7)	1.7 (±1.4)	0.02 (±0.02)	0.02 (±0.01)	Potato	3.1 (±2.5)	2.3 (±1.8)	0.03 (±0.02)	0.02 (±0.02)

**Table 4 nutrients-12-02079-t004:** Top 15 food group intake for Māori and non-Māori men low and adequate protein intake based on the Estimated Average Requirement (EAR).

Māori Men					Non-Māori Men				
	Contribution to Protein Intake (%)		Protein Intake (g/kg/day)			Contribution to Protein Intake (%)		Protein Intake (g/kg/day)	
Food Groups	Low protein (≤0.85 g/kg/day)	Adequate protein (>0.86 g/kg/day)	Low protein (≤0.85 g/kg/day)	Adequate protein (>0.86 g/kg/day)	Food Groups	Low protein (≤0.85 g/kg/day)	Adequate protein (>0.86 g/kg/day)	Low protein (≤0.85 g/kg/day)	Adequate protein (>0.86 g/kg/day)
	Mean (SD)	Mean (SD)	Mean (SD)	Mean (SD)		Mean (SD)	Mean (SD)	Mean (SD)	Mean (SD)
Fish, seafood	10.9 (±17.0)	11.2 (±17.5)	0.11 (±0.17)	0.11 (±0.18)	Beef, veal	11.1 (±13.3)	13.9 (±16.3)	0.11 (±0.13)	0.14 (±0.16)
Poultry	6.6 (±10.7)	10.4 (±16.9)	0.07 (±0.11)	0.10 (±0.17)	Bread	13.8 (±8.3)	11.0 (±6.2)	0.14 (±0.08)	0.11 (±0.06)
Beef, veal	12.0 (±18.2)	9.5 (±13.7)	0.12 (±0.18)	0.10 (±0.14)	Milk	12.6 (±8.9)	10.0 (±7.0)	0.13 (±0.09)	0.10 (±0.07)
Pork	6.8 (±11.0)	9.0 (±15.5)	0.07 (±0.11)	0.09 (±0.16)	Poultry	2.7 (±6.8)	9.8 (±14.9)	0.03 (±0.07)	0.10 (±0.15)
Bread	11.7 (±8.5)	8.8 (±5.7)	0.12 (±0.08)	0.09 (±0.06)	Fish, seafood	4.6 (±8.2)	8.0 (±11.8)	0.05 (±0.08)	0.08 (±0.12)
Milk	9.8 (±7.4)	7.5 (±5.8)	0.10 (±0.07)	0.07 (±0.06)	Cereals	6.4 (±4.3)	5.4 (±3.6)	0.06 (±0.04)	0.05 (±0.04)
Mutton (Sheep, lamb)	0.2 (±0.7)	5.9 (±14.7)	0.00 (±0.01)	0.06 (±0.15)	Pork	5.2 (±8.1)	4.3 (±8.9)	0.05 (±0.08)	0.04 (±0.09)
Cereals	5.5 (±4.2)	4.5 (±3.7)	0.06 (±0.04)	0.04 (±0.04)	Cheese	3.3 (±5.3)	4.1 (±5.3)	0.03 (±0.05)	0.04 (±0.05)
Egg, egg dishes	3.0 (±6.4)	4.1 (±5.1)	0.03 (±0.06)	0.04 (±0.05)	Vegetables	3.9 (±4.1)	3.8 (±2.5)	0.04 (±0.04)	0.04 (±0.02)
Sausage, processed meats	3.6 (±9.9)	3.3 (±8.2)	0.04 (±0.10)	0.03 (±0.08)	Egg, egg dishes	4.1 (±7.0)	3.2 (±4.7)	0.04 (±0.07)	0.03 (±0.05)
Vegetables	3.5 (±2.4)	3.3 (±2.6)	0.03 (±0.02)	0.03 (±0.03)	Potato	3.6 (±2.5)	3.0 (±1.9)	0.04 (±0.02)	0.03 (±0.02)
Potato	3.5 (±3.3)	3.0 (±1.8)	0.03 (±0.03)	0.03 (±0.02)	Mutton (Sheep, lamb)	1.0 (±4.5)	2.8 (±8.0)	0.01 (±0.04)	0.03 (±0.08)
Cheese	2.5 (±4.4)	2.1 (±4.1)	0.02 (±0.04)	0.02 (±0.04)	Cakes	1.9 (±3.6)	2.4 (±3.2)	0.02 (±0.04)	0.02 (±0.03)
Cakes	2.6 (±4.1)	1.9 (±2.8)	0.03 (±0.04)	0.02 (±0.03)	Pies, pasties	3.4 (±8.2)	2.2 (±5.2)	0.03 (±0.08)	0.02 (±0.05)
Non-alcoholic beverages	1.8 (±1.1)	1.8 (±2.3)	0.02 (±0.01)	0.02 (±0.02)	Sausage, processed meats	2.7 (±6.9)	2.0 (±5.3)	0.03 (±0.07)	0.02 (±0.05)

**Table 5 nutrients-12-02079-t005:** Reduced multivariate regression model predicts protein intake in grams per kilogram per day.

Effect	Gender	Estimate	Standard Error	*p* > (t)
**Intercept**		1.7720	0.8056	0.0283
**Age (Years)**		−0.01198	0.009609	0.2131
**Gender**	Men	0.09068	0.03057	0.0032
	Women	0	-	-
**Ethnicity**	Non-Māori	0.01965	0.03926	0.6169
	Māori	0	-	-
**Depression**	No	0.09467	0.04314	0.0287
	Yes	0	-	-
**SF-12 Physical Health Score**		0.001965	0.001450	0.1761

Reduced multivariate regression model was built to predict protein intake in g/kg/day. All variables not significant to *p*-value of <0.2 were removed except for age and ethnicity. Depression and being a woman were predicted to contribute to having lower protein intake in g/kg/day.

**Table 6 nutrients-12-02079-t006:** Logistic regression model to predict meeting the EAR for protein in grams per kilogram per day.

	Odds Ratio Estimates				
Effect		Point Estimate	95% Wald Confidence Limits		*p*-Value
**Age**	(one year greater)	1.079	0.943	1.234	0.2697
**Gender**	Men vs Women	0.759	0.484	1.191	0.2307
**Ethnicity**	Non-Māori vs Māori	1.360	0.764	2.418	0.2958
**Practical support available**	No vs. Yes	1.245	0.552	2.808	0.5972
**Depression**	No vs. Yes	1.415	0.726	2.756	0.3078
**Living alone**	No vs. Yes	0.958	0.610	1.502	0.8505
**NZ Deprivation score**	1–4 vs. 8–10	0.855	0.485	1.510	0.3806
	5–7 vs. 8–10	1.261	0.768	2.071	
**Wear dentures**	No vs. Full	1.335	0.763	2.336	0.0362
	Partial vs. Full	1.948	1.170	3.244	
**Swallowing problems**	Never vs. Often or always	0.835	0.245	2.855	0.8244
	Rarely vs. Often or always	0.924	0.208	4.101	
	Sometimes vs. Often or always	0.615	0.155	2.435	
**Chewing problems**	Never vs. Often or always	1.835	0.684	4.927	0.5193
	Rarely vs. Often or always	2.724	0.677	10.965	
	Sometimes vs. Often or always	1.599	0.528	4.842	
**SF-12 Physical Health Score**	(one point greater)	1.009	0.988	1.032	0.4004
**SF-12 Mental Health Score**	(one point greater)	1.003	0.975	1.031	0.8527
**Pension only income**	No vs. Yes	1.066	0.679	1.673	0.7818

Logistic regression model meeting protein intake based on Estimated Average Requirements (EAR) in g/kg/day. *p*-value of <0.2 was significant.

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
