# Peer review of "Dietary Protein Intake and Determinants in Māori and Non-Māori Octogenarians. Te Puāwaitanga o Ngā Tapuwae Kia Ora Tonu: Life and Living in Advanced Age: A Cohort Study in New Zealand"

_nutrients, 2020, doi:10.3390/nu12072079_

Round 1
Reviewer 1 Report
The article is based on previous results obtained in the Cohort Study in New Zealand (LiLACS NZ).
The authors do not indicate a nutritional interest in studying the food consumption of Maori versus non-Maori.
Is this because their eating habits and food preferences are different? It may be of interest to the reader who is not familiar with these ethnic groups to make some reference to the eating behaviour of both ethnic groups if this is addressed in the introduction.
The methodology, although previously published, is given enough information to be able to understand the process, although I have some doubts:
- What food composition database do you use? It is not referenced.
-Is it the official nutritional database of the country? It would be convenient to reflect this information in the part of Material and methods.
If Tables 3a and 3b have already been published, although it is indicated in the text (reference bibliography 8) it should be indicated in the table itself.
In these tables, the ordering of the food groups in the Maori and Non-Maori column appears to be a function of Contribution to protein intake (%) in relation to adequate protein intake. is this true? In this case, the authors should indicate in the text the criteria followed to make this ranking and avoid confusion to the reader because the food groups have not been aligned by rows.
In the distribution of proteins in the meals, the authors do not discuss or look for an explanation because it is higher in the dinner. Can it be due to the food that is usually consumed at that meal?
Author Response
Comments and Suggestions for Authors
The article is based on previous results obtained in the Cohort Study in New Zealand (LiLACS NZ).
The authors do not indicate a nutritional interest in studying the food consumption of Maori versus non-Maori.
Is this because their eating habits and food preferences are different? It may be of interest to the reader who is not familiar with these ethnic groups to make some reference to the eating behaviour of both ethnic groups if this is addressed in the introduction.
The methodology, although previously published, is given enough information to be able to understand the process, although I have some doubts:
- What food composition database do you use? It is not referenced.
-Is it the official nutritional database of the country? It would be convenient to reflect this information in the part of Material and methods.
If Tables 3a and 3b have already been published, although it is indicated in the text (reference bibliography 8) it should be indicated in the table itself.
In these tables, the ordering of the food groups in the Maori and Non-Maori column appears to be a function of Contribution to protein intake (%) in relation to adequate protein intake. is this true? In this case, the authors should indicate in the text the criteria followed to make this ranking and avoid confusion to the reader because the food groups have not been aligned by rows.
In the distribution of proteins in the meals, the authors do not discuss or look for an explanation because it is higher in the dinner. Can it be due to the food that is usually consumed at that meal?
Submission Date 11 June 2020 Date of this review 02 Jul 2020 21:44:28
Response to Comments and Suggestions: Reviewer 1
The article is based on previous results obtained in the Cohort Study in New Zealand (LiLACS NZ). The authors do not indicate a nutritional interest in studying the food consumption of Maori versus non-Maori. Is this because their eating habits and food preferences are different? It may be of interest to the reader who is not familiar with these ethnic groups to make some reference to the eating behaviour of both ethnic groups if this is addressed in the introduction.
Response:
Thank you for your comment. The apparent lack of nutritional interest in studying the food consumption of Maori versus non-Maori warrants explanation. In Wave 1 (baseline)of LILACS NZ we investigated the nutrition risk status of Māori and non-Māori where half (49%) of Māori and 38% of non-Māori octogenarians were at high risk of malnutrition (reference 3). At follow up (Wave 2) a detailed examination of macronutrient (and micronutrient) intake was undertaken in 216 Māori and 362 non-Māori using a repeat 24hMPR (reference 8). As the picture of health for Māori is a reflection of their life course and the effects of social, economic and cultural deprivation from the ongoing process of colonisation and globalisation which is occurring and potentially affecting all domains of health status we do not compare and contrast the dietary intakes of Māori versus non-Māori. A kaitiaki group of Māori elders act as guardians to our participants and Māori values are respected. In Wave 2 Reference 8 we reported clear ethnic differences for energy and macronutrient intake and where culturally related food patterns, BMI and BMR differed. In Wave 1 reference 3 we reported disability affects 24% of Māori compared to 18% of non- Māori aged over 65 years (Ministry of Health 2011). This too may have an adverse effect on meal preparation, consumption and nutrient intake (Sharkey, Branch et al. 2002). Māori participants with only a primary level of education were three times more likely to be at high nutrition risk than those with a tertiary education. Across all socio-economic indicators including education older Māori are more disadvantaged than older non- Māori (Statistics New Zealand 2007). Nearly half of our Māori participants lived in the highest quintile of socio-economic deprivation. Older single Māori, mostly women, have the least material wellbeing (Cunningham, Durie et al. 2002) which may lead to low food security (University of Otago and Ministry of Health 2011).
The methodology, although previously published, is given enough information to be able to understand the process, although I have some doubts:
- What food composition database do you use? It is not referenced.
-Is it the official nutritional database of the country? It would be convenient to reflect this information in the part of Material and methods.
Response
Thank you. We have added details of the New Zealand Food Composition Database to the methods section as reported in the Macronutrient intake paper (reference 8). i.e. Nutrient intakes were calculated by coding all foods and drinks recorded by the participants using the New Zealand Food Composition Database (NZFCDB). FOODFiles (2010), an electronic subset of data from the NZFCDB was used as the main source of food composition data and contained information on 2739 foods.
If Tables 3a and 3b have already been published, although it is indicated in the text (reference bibliography 8) it should be indicated in the table itself.
Response
Thank you. In reference 8 we reported the % of contribution from food groups to energy and macronutrient intake for Māori and non-Māori by sex as bar graphs showing the foods sources contributing to ≥75% energy and macronutrients including protein. Tables 3a and 3b are presented by ethnicity and sex by low and adequate protein intake; for % contribution to protein intake and protein intake g/kg/d.
In these tables, the ordering of the food groups in the Maori and Non-Maori column appears to be a function of Contribution to protein intake (%) in relation to adequate protein intake. is this true? In this case, the authors should indicate in the text the criteria followed to make this ranking and avoid confusion to the reader because the food groups have not been aligned by rows.
Response
Thank you. You are correct the ordering of food groups was by Contribution to protein intake (%) in relation to adequate protein intake. This has now been added to the text. i.e As reported previously [8], % contribution from food groups to protein intake is shown in Tables 3a and 3b. Food groups are ranked from high to low by adequate protein intake (g/kg/d).
In the distribution of proteins in the meals, the authors do not discuss or look for an explanation because it is higher in the dinner. Can it be due to the food that is usually consumed at that meal?
Response
Yes I think we can make the assumption that protein is mostly consumed at dinner. We have added to section 4.2 protein distribution in the discussion “This may reflect the typical New Zealand eating pattern where protein is eaten mostly at the dinner meal”.
Reviewer 2 Report
This review by Ram et al. documents the quantity, distribution, and sources of dietary protein in the diets of Māori and non-Māori octogenarian men and women. They also investigated determinants of inadequate protein intake from various available outcomes. The methods are well-written and the results clearly communicated in the text and tables. Documenting the dietary protein intake and distribution patterns, particularly among aging adults, is important to the field. While this has occurred primarily in North American, European, and Asian cohorts, this type of research is needed in all cohorts. Therefore, this study will be a useful addition to the literature, documenting inadequate and unbalanced protein intakes exists among older adults across the globe. I only have a few minor comments and suggestions:
Minor comments:
Introduction:
- Second paragraph, last sentence: Please add a reference to this statement. The EAR and RDI for New Zealand is higher than in other countries, which may not be as well known. A citation, or perhaps text, indicating these values are based on the New Zealand Nutrient Reference Values could be helpful to the reader.
Methods:
- How did you handle implausible 24 h recall data? As we know, documenting dietary intake is an incredibly difficult process and can be unreliable. You appropriately indicate as much in the methods with reference to protein intake being more reliable. However, implausible reporting can occur. Did look at this possibility? If yes, how did you handle these data? One method is to reference a minimum and maximum REE for a given participant and exclude data below and above this point.
Discussion:
- Section 4.1, first paragraph: You compare the results from this study to the Newcastle study; however, the EAR is the estimated intake for 50% of the population. The RDA, used in the Newcastle study, is for 2 SD above the EAR. Therefore, the prevalence of low protein intake among adults in the Newcastle study is likely lower if they used the EAR as the cutoff. The percentage in this study is likely higher if you used the RDI. Perhaps it would be better to use g/kg/d cutoffs here to discuss the proportions consuming an inadequate protein intake than using the EAR and RDI/RDA designations? I understand you discussed the EAR and RDI here because they are similar in g/kg/d; however, it makes it confusing when those are the main focus, rather than the g/kg/d values.
- You mention a third of participants did not meet the EAR. Do you think this is significant? By definition, the EAR only needs to be met by half the population. For your cohort, two-thirds meet this definition. It may be worth noting this with the disclaimer that a sample distribution should not be mistaken for the population distribution. That is, you cannot know if the participants consuming a lower protein diet are the ones who require less protein. This could be partially answered with the following suggestions:
Suggestions:
- In the methods, you indicate that handgrip strength was measured in addition to the 24 h MPR. There is considerable interest in the relationship between dietary protein intake quantity and distribution with body composition and physical performance outcomes (1). I suggest you relate protein quantity and distribution to these outcomes. This would be a considerable addition to the paper that would increase its relevance. There are considerable ways to quantity protein distribution. See the review below for different methodological examples from observational studies on protein distribution (1). 1) You could create a protein distribution CV throughout the day. 2) Look at the number of meals that reach a target threshold. E.g. 0.4 g/kg or 30 g of protein. 3) Look at what happens when at least one meal does not reach a target threshold. You could related this to grip strength and body composition outcomes.
- Hudson, J.L.; Bergia, R.E., III; Campbell, W.W. Protein Distribution and Muscle-Related Outcomes: Does the Evidence Support the Concept? Nutrients 2020, 12, 1441
- Please consider looking at the protein distribution patterns of participants that are in different protein intake categories as well. There is interest in knowing by what method are/can older adults achieve higher protein diets. They are likely having higher protein breakfast. Indeed, considerable research is being done on breakfast consumption to improve protein intake/distribution. This study could contribute to this literature base by stratifying by total daily intake into tertiles, quantiles, or quintiles and identifying in which meals protein intake goes up and if protein distribution becomes more balanced.
- In the conclusions, you state the protein distribution was unbalanced. I do not disagree with this conclusion. Clearly, intakes than 2x and 3x greater at dinner than breakfast indicate an unbalanced distribution. However, this is subjective. Perhaps there is an opportunity here to objectively indicate distribution is unbalanced? Or to operationalize a definition. It may be as simple as running an ANOVA to indicate protein quantity per meal is different.
- There are a considerable number of protein distribution-related articles that were not cited. While I’m not advocating for an exhaustive citation of the literature, consider reviewing the existing literature again for relevant articles in the case some were missed in the review process
Author Response
Response to Comments and Suggestions: Reviewer 2
Thank you for the helpful comments and suggestions
This review by Ram et al. documents the quantity, distribution, and sources of dietary protein in the diets of Māori and non-Māori octogenarian men and women. They also investigated determinants of inadequate protein intake from various available outcomes. The methods are well-written and the results clearly communicated in the text and tables. Documenting the dietary protein intake and distribution patterns, particularly among aging adults, is important to the field. While this has occurred primarily in North American, European, and Asian cohorts, this type of research is needed in all cohorts. Therefore, this study will be a useful addition to the literature, documenting inadequate and unbalanced protein intakes exists among older adults across the globe. I only have a few minor comments and suggestions:
Minor comments:
Introduction:
- Second paragraph, last sentence: Please add a reference to this statement. The EAR and RDI for New Zealand is higher than in other countries, which may not be as well known. A citation, or perhaps text, indicating these values are based on the New Zealand Nutrient Reference Values could be helpful to the reader.
Response
Thanks you for pointing this out. A reference is added as follows: Adequate protein intake is important for optimal muscle synthesis wherewith protein intake below the estimated average requirements (EAR) of 0.75g/kg/d for women and 0.86g/kg/d for men using the New Zealand Nutrient Reference Values (NRVs) [25] can lead to decreased physical function and can contribute to sarcopenia.
Methods:
- How did you handle implausible 24 h recall data? As we know, documenting dietary intake is an incredibly difficult process and can be unreliable. You appropriately indicate as much in the methods with reference to protein intake being more reliable. However, implausible reporting can occur. Did look at this possibility? If yes, how did you handle these data? One method is to reference a minimum and maximum REE for a given participant and exclude data below and above this point.
Response
Thank you we did report the sensitivity analysis in the macronutrient intake paper (reference 8) where we excluded those with EI:BMRest <0.9 and EI:BMRest >2 and presented this data as online supplementary Tables S5 and S6.
Discussion:
- Section 4.1, first paragraph: You compare the results from this study to the Newcastle study; however, the EAR is the estimated intake for 50% of the population. The RDA, used in the Newcastle study, is for 2 SD above the EAR. Therefore, the prevalence of low protein intake among adults in the Newcastle study is likely lower if they used the EAR as the cutoff. The percentage in this study is likely higher if you used the RDI. Perhaps it would be better to use g/kg/d cutoffs here to discuss the proportions consuming an inadequate protein intake than using the EAR and RDI/RDA designations? I understand you discussed the EAR and RDI here because they are similar in g/kg/d; however, it makes it confusing when those are the main focus, rather than the g/kg/d values.
Response
Thank you this is a valid point and we have altered the text accordingly. The sentence now reads “This prevalence of low protein intake is similar to findings among adults (85 years) in the Newcastle 85+ study where the median protein intake was 0.96g/kg/d women and 1.04 g/kg/d for men28% of participants consumed a protein intake less than the UK RDA of <0.8g/kg/d [10]”.
- You mention a third of participants did not meet the EAR. Do you think this is significant? By definition, the EAR only needs to be met by half the population. For your cohort, two-thirds meet this definition. It may be worth noting this with the disclaimer that a sample distribution should not be mistaken for the population distribution. That is, you cannot know if the participants consuming a lower protein diet are the ones who require less protein. This could be partially answered with the following suggestions:
Suggestions:
- In the methods, you indicate that handgrip strength was measured in addition to the 24 h MPR. There is considerable interest in the relationship between dietary protein intake quantity and distribution with body composition and physical performance outcomes (1). I suggest you relate protein quantity and distribution to these outcomes. This would be a considerable addition to the paper that would increase its relevance. There are considerable ways to quantity protein distribution. See the review below for different methodological examples from observational studies on protein distribution (1). 1) You could create a protein distribution CV throughout the day. 2) Look at the number of meals that reach a target threshold. E.g. 0.4 g/kg or 30 g of protein. 3) Look at what happens when at least one meal does not reach a target threshold. You could related this to grip strength and body composition outcomes.
- Hudson, J.L.; Bergia, R.E., III; Campbell, W.W. Protein Distribution and Muscle-Related Outcomes: Does the Evidence Support the Concept? Nutrients 2020, 12, 1441
- Please consider looking at the protein distribution patterns of participants that are in different protein intake categories as well. There is interest in knowing by what method are/can older adults achieve higher protein diets. They are likely having higher protein breakfast. Indeed, considerable research is being done on breakfast consumption to improve protein intake/distribution. This study could contribute to this literature base by stratifying by total daily intake into tertiles, quantiles, or quintiles and identifying in which meals protein intake goes up and if protein distribution becomes more balanced.
- In the conclusions, you state the protein distribution was unbalanced. I do not disagree with this conclusion. Clearly, intakes than 2x and 3x greater at dinner than breakfast indicate an unbalanced distribution. However, this is subjective. Perhaps there is an opportunity here to objectively indicate distribution is unbalanced? Or to operationalize a definition. It may be as simple as running an ANOVA to indicate protein quantity per meal is different.
- There are a considerable number of protein distribution-related articles that were not cited. While I’m not advocating for an exhaustive citation of the literature, consider reviewing the existing literature again for relevant articles in the case some were missed in the review process
Response
Thank you for the suggestions. We agree we cannot know if the participants consuming a lower protein diet are the ones who require less protein. We have continued to investigate outcomes of protein intake and have a manuscript in preparation related to grip strength and body composition as you suggest. Thank you for the points you have raised, we will endeavour to address these.